# The Fall Risk Screening Scale Is Suitable for Evaluating Adult Patient Fall

**DOI:** 10.3390/healthcare10030510

**Published:** 2022-03-11

**Authors:** Li-Chen Chen, Yung-Chao Shen, Lun-Hui Ho, Whei-Mei Shih

**Affiliations:** 1Department of Nursing, Linkou Chang Gung Memorial Hospital, Taoyuan 333, Taiwan; a22059@cgmh.org.tw (L.-C.C.); ho1180@cgmh.org.tw (L.-H.H.); 2Department of Nursing, Chang Gung University of Science and Technology, Taoyuan 333, Taiwan; q12185@cgmh.org.tw; 3Department of Nursing, New Taipei Municipal Tucheng Hospital Built and Operated by Chang Gung Medical Foundation, New Taipei City 236, Taiwan; 4Graduate Institute of Gerontology and Health Care Management, Chang Gung University of Science and Technology, Taoyuan 333, Taiwan

**Keywords:** fall risk assessment, screening tool, patient falls, patient safety

## Abstract

(1) Background: This study aimed to test the feasibility of utilizing the screening tool for fall risk assessment in adult inpatient and verify its accuracy in a medical center in Taiwan. (2) Methods: This study retrospectively collected all adult fall cases among inpatients occurring in the general wards of a medical center between 1 January 2013 and 31 December 2015. This inpatient fall risk screening scale was measured by the sensitivity, specificity, and accuracy. (3) Results: There were 1331 (0.4%) falls among a total of 357,395 inpatients during this period. Factors predictive of falling risk included: age, consciousness, body shift assistance, use of fall risk medications, fall history, dizziness or weakness, toileting, and impaired mobility. Using the eight-factor assessment, two was the best cutoff point for identifying the fall risk group, with area under Receiver Operating Characteristic (ROC) curve (AUC) = 0.817, sensitivity = 80.93%, specificity = 73.0%, accuracy = 73.03%, and likelihood ratio = 11.48. (4) Conclusions: The accuracy of the eight-item fall risk assessment tool created for this study was validated. These results can serve as a reference for institutions to develop more effective fall risk assessment scale for inpatients, enabling clinical nurses to identify and more comprehensively assess the groups at highest risk for falling during their hospital stay.

## 1. Introduction

According to Joint Commission of Taiwan, prevention of patient fall is one of the national patient safety goals [1]. Surveys reveal that inpatient falls are the third most frequently occurring of all hospital accidents [2]. In less serious cases, inpatient falls may not cause injuries, but in severe cases, they can lead to serious complications, prolong the length of hospital stays, and increase the costs of medical care [3,4]. The U.S. Joint Commission has been advocating patient-centered care since 1999 to promote patient safety. In 2007, the U.S. Joint Commission recommended that all medical institutions establish a standard assessment program for reducing patient falls, regularly re-evaluate patients’ risk of falling, and take measures to prevent falls and improve their track records [5]. The Joint Commission of Taiwan has likewise included the prevention of patient falls and reduction of injuries among the annual objectives for improving patient safety and quality of care at medical institutions. Commission guidelines emphasize implementation of fall risk assessment and precautionary measures, use of reliable risk assessment scale, early detection of patients at high risk of falling, as well as regular review of the effectiveness of assessment procedures, methods and scales for each hospital. According to the Taiwan patient safety reporting system (TPR), there were 14,308 hospital falls in 2020, 70.9% [2] of them among patients assessed as at high risk of falling, revealing the importance of fall risk screening tool.

Injuries resulting from inpatient falls can cause extra financial burden and decreased revenue to healthcare facilities. On average, injurious falls lead to an increased length of hospital stay by 6 to 12 days and an additional cost of $13,316, and that fallers stayed 6.3 days longer than non-fallers at three hospitals in a Midwestern health care system in US [3]. Similar results were also found in Australia that patients wo experience an in-hospital fall have significantly longer hospital stays and higher cost [4]. An unexpected and significant decline of 2 to 3 min per hour in the time nurses spent on fall-related activities equated to a projected gross savings of $0.8 million to $1.9 million per hospital per year [6]. In 2011, there were 77,086 cases of falls which required hospitalization in the United States, and almost 80% of them were among adults over 50 years of age. Total medical costs associated with falling increased from US $1.9 billion in 2007 to US $3.1 billion in 2011 [7]. According to Netherland’s report, from 2000 through 2016, an increase in the total number of deaths from falls in Dutch persons 80 years and older was from 391 deaths in 2000 to 2501 in 2016). The overall crude mortality rate per 100 000 population increased from 78.1 (95% CI, 70.4–85.9) in 2000 to 334.0 (95% CI, 320.9–347.1) in 2016 (*p* < 0.001) [8]. These figures show that prevention of falls is an important for reducing healthcare expenditures and improving patient care in acute care medical institutions.

Falls are not caused by a single factor. Fall risk factors proposed by many studies include: being male, between 15 and 65 years of age, having a history of falls, persistent or intermittent cognitive disabilities, dizziness, weakness, frequent urination, diarrhea, requiring assistance in the toilet, requiring assistive devices, unstable blood pressure, being underweight, arthritis, diabetes, insomnia, depression, being unable to get up or walk, being unaccompanied during hospitalization, taking multiple medications, lower limb weakness, poor balance, vision or hearing loss, cognitive dysfunction, gait disorder, and daily life dysfunction [9,10,11,12,13,14,15,16]. Different research have marked explanations of various risk factors, and there is no gold standard of fall risk assessment scales.

Our hospital routinely performs fall risk factor assessment for hospitalized patients within eight hours of admission. At the time of this study, the high risk factors assessed included: age, consciousness, use of oxygen, body shift assistance, use of fall risk medications, fall history, and dizziness or weakness. If a patient had a score of three points or higher on the fall risk factor assessment, our hospital immediately established a nursing plan. Even so, 513 inpatient falls occurred in 2015. Among these, 11% of patient falls were due to frequent toileting and 43.2% were due to impaired mobility and yet these two risk factors were not in our current screening tool. According to Taiwan patient safety net, Ministry of Health and Welfare of Taiwan and the most popular and the greatest diagnostic value screening tool- STRATIFY, frequent toileting and impaired mobility are included in the risk factors assessment [1,17]. Based on the facts of data and literature review, we decided to include these two factors into the fall risk screening scale. An accurate and quick fall risk screening tool in identifying patients at high risk of falling in the hospital is the most important step in preventing falls and avoiding their consequences. The lack of accurate and consistent patient falls risk assessments, the risk of more severe falls increased [18]. Therefore, the purpose of this study was to revise the original tool in order to create a more accurate inpatient fall risk screening tool.

## 2. Materials and Methods

### 2.1. Research Design

Research data downloaded from the HIS (Hospital Information System) in this research hospital included the correlation between inpatient falling and fall risk assessment as well as the correlation between inpatient falls and patient’s health evaluation results.

### 2.2. Setting and Sample

This study retrospectively collected all adult fall cases among inpatients occurring in the general wards of a northern medical center in Taiwan between 1 January 2013 and 31 December 2015 (Figure 1).

### 2.3. Research Instrument

In addition to conducting descriptive statistical analysis, this study analyzes the “summary table of high fall risk factors assessment for adults”, “high fall risk factors”, and the basic information of inpatients that fell, in order to explore the correlation between high fall risk status and the actual occurrence of falls.

This study used inpatient falls as a dependent variable, with the independent variables as follows: age, consciousness, use of oxygen, body shift assistance, use of fall risk medications, fall history, dizziness or weakness, toileting, and impaired mobility. Since data was collected based on “yes” or “no”, the logistic regression was used for binary variables [19].

### 2.4. Data Analysis

In order to test a good tool, sensitivity, specificity, and associated confidence interval and ROC analysis should be implemented. Sensitivity evaluates how good the test is at detecting a positive disease. Specificity estimates how likely patients without disease can be correctly ruled out. LRs allow providers to determine how much the utilization of a particular test will alter the probability. ROC curve is a graphic presentation of the relationship between both sensitivity and specificity and it helps to decide the optimal model through determining the best threshold for the diagnostic test [20]. Therefore, we analyzed the data using the SPSS 22.0 software (IBM, Armonk, NY, USA) suite and inspected the fall risk factors using a receiver operating characteristic curve (ROC curve) so as to analyze their accuracy and calculate tangency points. This study also calculated the sensitivity, specificity, positive and negative predictive rate, accuracy and likelihood ratio for each risk factor. It also verified the odds ratio to predict risk factors using logistic regression, with the level of significance set at *p* < 0.05.

## 3. Results

### 3.1. Attribute of the Subjects and Fall Conditions

There were 1331 falls out of 357,396 inpatients, accounting for 0.37% of hospitalized patients. The average score on the fall risk assessment for patients who fell was two.

### 3.2. Accuracy and Optimal Tangent Point for a Tool to Assess Inpatient Fall Risk Factors

This study analyzed the high fall risk factor assessment items in relation to the results of physical assessment using univariate analysis. The fall risk factors included in this new tool included “age”, “consciousness”, “body shift assistance”, “use of fall risk medications”, “fall history”, “dizziness or weakness”, “toileting”, and “impaired mobility”. Heart rate, respiration rate, peristalsis and oxygen use were eliminated. The results of logistic regression analysis showed that eight of the fall risk factors had significant predictive power (Table 1).

This study then examined the accuracy, optimal tangent point, sensitivity and specificity of the newly constructed fall risk factor scale. The area under ROC curve (AUC) was used to evaluate the discriminating power of the assessment scale. When the AUC is between 0.5 and 1, the closer it is to 1, the greater the accuracy. When the AUC is between 0.5 and 0.7, the accuracy is lower. An AUC is between 0.7 and 0.8 indicates moderate accuracy. AUC > 0.9 represents the highest accuracy. We identified the best tangent point in accordance with Yunden’s index as follows: sensitivity + specificity -1. When the resulting number is between 0 and 1, the closer it is to 1, the better the sensitivity and specificity. At the same time, likelihood ratios (LR) of sensitivity and specificity can help evaluate the accuracy of the test. When LR > 10, that indicates the tool in question has great empirical clinical significance. When LR is between two and five, the tool has less significance for clinical judgment [21].

The results revealed that the best tangent point for the new, eight-item fall risk factor assessment scale is 1.5 points (Table 2). Further we examined the predictive accuracy of this new fall risk assessment scales. For the new eight-item fall risk scale, a score of two points was the best tangent point for identifying the high fall risk group, while the AUC = 0.817, sensitivity = 80.93%, specificity = 73.0%, accuracy = 73.03%, and the likelihood ratio = 11.48. According to Sardanelli & Di Leo [22], the probability of falling increases when the likelihood ratio is over 10 points.

In summary, using the eight-factor assessment, two was the best cutoff point for identifying the fall risk group, with area under ROC curve (AUC) = 0.817, sensitivity = 80.93%, specificity = 73.0%, accuracy = 73.03%, and likelihood ratio (LR) = 11.48. It indicated that the new scale created for this study in this institution is a reliable fall risk scale based on diagnosis test statistics model [20]. (Figure 2, Table 3).

## 4. Discussion

Institutions select fall risk factors based on the nature of their inpatient population, severity of disease and literature review. This study employed multivariate analysis to analyze this research institution’s inpatient fall risk assessment factors, evaluating “age”, “consciousness”, “body shift assistance”, “use of fall risk medications”, “fall history”, “dizziness or weakness”, “toileting”, and “impaired mobility”. Results showed that these eight fall risk factors are related to incidents of falling. Multivariate regression analysis revealed that all eight fall risk assessment factors had significant predictive power.

A review of fall risk factors mentioned in the literature shows that many scholars have concluded that age has significant power for predicting falls which is similar to our result [13,15]. In terms of “consciousness” as a variable, our results show that disturbance of consciousness does have significant power for predicting falls, which is consistent with findings of other studies [13,14]. Our study also found that “body shift assistance” has significant power for predicting falls, which is likewise consistent with the findings of most scholars [13,14,15,23]. We likewise found that “use of fall risk medications” had significant power to predict falls, in line with findings of many researchers [13,14,15]. A “history of fall history” also had predictive power for falls, consistent with other studies’ findings [5,14]. In agreement with other studies, we found the variables “dizziness or weakness” had significant predictive power [14,15]. Likewise, two more factors proposed by this study—“toileting” and “impaired mobility”—showed significant predictive power for falls, consistent with the research findings of scholars such as Lin et al. and Tinetti & Kumar [13,24].

This study developed a fall risk screening scale with eight factors by searching related literature and analyzing risk factors and health assessment data for inpatients who fell and for all inpatients at a single medical center over a 3 year span. The sensitivity of the scale with 80.93% and the specificity with 73.0% is better than the old one with 66.9% sensitivity and 68.4% specificity as well as nearby countries such as Japan with 71.3% sensitivity and 66.0% specificity, and Korea with 78.9% sensitivity and 55.8% specificity respectfully [25,26]. The likelihood ratio of 11.48 and the overall prediction accuracy of 73.03% are good. This eight-factor fall risk screening scale set a score of ≥2 for identifying high fall risk patients is suitable for this institution.

By analyzing the results of health assessments conducted before patients fell, this study found frequent toileting to be the most powerful predictive factor. Frequent toileting had a 4300% prediction rate for falls, and impaired mobility had a prediction rate of 300%, therefore we recommend that these two items have additive percentage on scores of fall risk factors. Our results suggest it would be better to use the new version of the fall risk factor assessment scale. “Age,” consciousness,” “body shift assistance”, “use of fall risk medication”, “fall history”, and “dizziness or weakness”, each counted for one point. However, “toileting” and “impaired mobility”, each counted for two points. A total score of two points indicates a high fall risk patient; and nurses should then employ individualized and appropriate fall prevention measures. Due to many factors that lead to falls, there is no gold standard tool that can be used in performing a perfect fall risk assessment. For this reason, a simultaneous application of multiple tools is recommended, and an in-depth interview by the healthcare professionals is essential. In addition, we recommend that an initial assessment be carried out when the patient is first admitted to the hospital. If daily health assessment of the patients reveals the patient has frequent urination and weakness, a re-assessment of the patient’s fall risk status should be carried out.

The new fall risk screening scale was then introduced in 2018. Before this fall risk screening scale was implemented, in-house nursing staff training was held to promote the accuracy use of this tool. We simultaneously activated patient fall prevention monitoring system. Since this screening tool was implemented, the numbers of patient falls dropped from 513 (0.49%) in 2015 with the old screening tool to 218 (0.21%) in 2018 and 206 (0.20%) in 2019 indicating the new tool can be accurately used for clinical nurses to identify and more comprehensively assess the groups at highest risk for falling during their hospital stay.

## 5. Conclusions

The causes contributing to patient falls are complex. After analyzing data from 357,396 inpatients, this study determined that the prediction accuracy of the previous fall risk factor scale used by the research institution was weak, and that the factor “use of oxygen” in particular, was not significant. Based on analysis of the inpatient data, this study proposed using eight factors for fall risk assessment. These include “age”, “consciousness”, “body shift assistance”, “use of fall risk medications”, “fall history”, “dizziness or weakness”, “toileting”, and “impaired mobility”.

Although this study was taken place in a medical center, our findings can serve as an empirical reference for amending patient safety policies and for implementing measures that should prevent the occurrence of inpatient falls, or at least reduce the severity of injuries. Based on the results of this study, it is recommended that nurses should have keen observations when caring patient for screening fall risk factors and compare institution’s tool and literature so as to come up with research questions in the future study.

In conclusion, this patient fall screening scale is suitable for evaluating adult patient fall.

## Figures and Tables

**Figure 1 healthcare-10-00510-f001:**
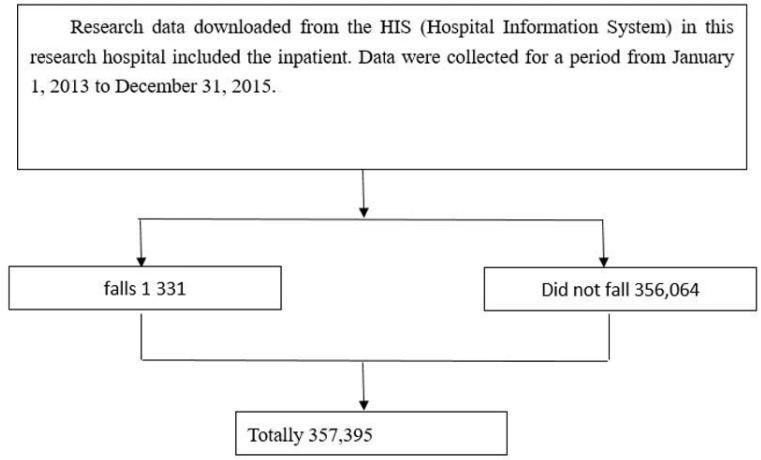
Flowchart of study subjects’ selection.

**Figure 2 healthcare-10-00510-f002:**
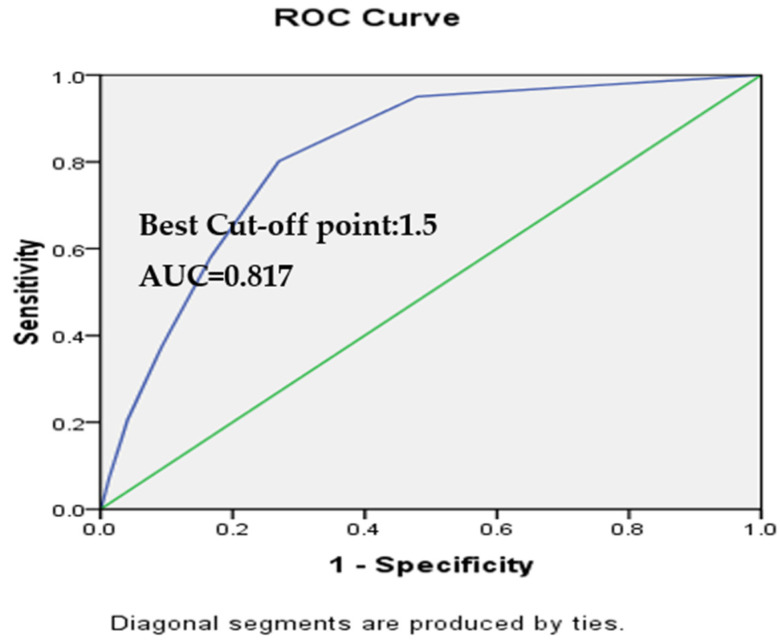
ROC Curve of New version fall risk assessment scale.

**Table 1 healthcare-10-00510-t001:** Logistic Regression Analysis of Fall Risk Assessment Factors.

Risk Factors	Fall Risk Assessment Factors						
	Esti-Mated Value of B	S.E.	Wald	*p* Value	Exp(B)	EXP(B) 95.0% Confidence Interval	
						Lower Bound	Upper Bound
Age	0.377	0.061	39.130	<0.001	1.458	1.296	1.641
Consciousness	0.716	0.079	82.092	<0.001	2.047	1.753	2.390
Body shift assistance	−0.859	0.076	127.756	<0.001	0.424	0.365	0.492
Use of fall risk medications	0.284	0.062	21.075	<0.001	1.328	1.177	1.499
Fall history	0.344	0.087	13.730	<0.001	1.410	1.190	1.671
Dizziness or weakness	0.264	0.067	15.371	<0.001	1.303	1.141	1.487
Toileting	3.767	0.063	3570	<0.001	43.239	38.214	48.926
Impaired mobility	1.340	0.073	337.728	<0.001	3.819	3.311	4.406

*p* < 0.001. (note: Estimated value of B—Estimated value of coefficients (Beta); S.E.—standard error; Wald—a way to find out if explanatory variables in a model are significant; *p* value—test whether it is significant or not; Exp(B)-Standardized regression coefficients (Beta); EXP(B) 95.0% confidence interval- test the true mean value (μ)).

**Table 2 healthcare-10-00510-t002:** The Best Tangent Point for Fall Risk Factor Assessment Scale Score.

Tangent Point	Sensitivity	1-Specificity	Yunden’s Index
−1	1.00	1.00	0.00
0.5	0.950	0.479	0.471
**1.5**	0.802	0.270	**0.532 b**
2.5	0.582	0.167	0.415
3.5	0.376	0.093	0.283
4.5	0.204	0.040	0.164
5.5	0.074	0.013	0.061
6.5	0.017	0.003	0.014
7.5	0.002	0.000	0.002
9.0	0.000	0.000	0.00

Note: **b**—best of cut-off point.

**Table 3 healthcare-10-00510-t003:** Accuracy Analysis of the best tangent point of the fall risk assessment scale. *n* = 357,396.

Dependent Variable		Not High Risk Cases ≤ 1	High Risk Cases ≥ 2	Total
Fall	No	259,929	96,135	356,064
	Yes	254	1078	1332
Total		260,183	97,213	357,396
AUC, 95% CI		0.817, 95% CI (0.808–0.827)		
Sensitivity		80.93		
Specificity		73.00		
Prediction rate of positive		1.11		
Prediction rate of negative		99.90		
Accuracy		73.03		
Likelihood ratios		11.48		

## Data Availability

The data and the questionnaires of the study are available upon request from the corresponding author.

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
