# Peer review of "The Fall Risk Screening Scale Is Suitable for Evaluating Adult Patient Fall"

_healthcare, 2022, doi:10.3390/healthcare10030510_

Round 1

Reviewer 1 Report

The authors have presented a detailed statistical analysis on factors that may be used to assess fall-risk for inpatients. The use of data from hospital to validate the developed assessment tool is helpful in validating the strategy. Following are few comments that I have on the manuscript:

  • Sensitivity of the current TPR system is 70.9% and with this new tool, it improves to 80.9%. Is that correct? What would be practical challenges and pitfalls in moving from the current TPR to this new tool with additional factors?

  • For a multivariate analysis, are these factors assumed to be independent? Has any analysis been performed on correlations between these?

  • Why use the logistic regression model which is linear in nature? Would a comparative study with other modelling strategies yield different (better?) results?

  • Can the logistic regression analysis be explained in further detail? For example, no information about terms (B, Wald, S.E. etc.) used in Table 1 is presented.

  • How is class imbalance handled in this analysis?

  • Why is age highlighted (bold) in Table 1?

  • The conclusions section presents new results with prospective validation on use of this tool. This should be explained in more detail and added likely to the results section before being discussed in the paper.

  • The data from 2015, 2018 and 2019 are 0.48%, 0.021% and 0.020%. Is this correct? While the absolute numbers have reduced from 513 to 206, the relative values have decreased from 0.48% to 0.020%. Is this a typo or base effect?

Reviewer 2 Report

Introduction: It has been explained why a study should be done for "Fall risk factor assessment", but “why this is important” is not explained sufficiently according to the international literature. This should be explained with the support of the literature. How are other countries/hospitals examples? Does the International Medical Association (for this field) have protocols, and if so, how? Questions such as: Because this constitutes the rationale of the research. In addition, it should be explained why this research is "original".

Method: Why these statistical techniques are preferred should be briefly explained based on "textbook/s” and other literature.

Discussion: There has been enough discussion, but the originality of the research has not been adequately interpreted. For example, the answer to the question "How can the results of this study be compared with the results of other studies to be uniquely defended?" should be included here. This should reflect the analytical thinking powers of the researchers.

Conclusion: An explanation has been made regarding the use of the results. Although this is valuable, it should be stated what research other researchers can do in the future based on the results of this research. That is, research questions and topics should be suggested.

Round 2

Reviewer 1 Report

Thank you for the revised version. I have a few minor comments:

  • Define ROC when it is used for the first time in the text as well as in the abstract.
  • The data from 2018 and 2019 is only introduced in the conclusions section of the manuscript. Rest of the manuscript is based on data collected till 2015. Prospective validation results with discussion could be included in the results or discussion section and not be introduced for the first time in the conclusions section.
  • Number of falls with new screening reduced from 513 in 2015 to 218 in 2018. This according to the paper in relative terms is 0.49% and 0.02% of total patients. How did reducing the falls by approximately a factor of 2 reduce the relative value by approximately 20 times? Is this a consequence of base effect, i.e., did the number of patients in the hospital increase 10 fold over three years?
  • The new text added includes minor typos and errors. Kindly proof read and correct these errors.

Reviewer 2 Report

Most of the suggested fixes have been made. The rationale and conclusion part of the research has been enriched. The discussion section could be enriched a little more.
